# LANGUAGE-DRIVEN OPEN-VOCABULARY KEYPOINT DETECTION FOR ANIMAL BODY AND FACE

## ABSTRACT

Current approaches for image-based keypoint detection on animal (including human) body and face are limited to specific keypoints and species. We address the limitation by proposing the **O**pen-**V**ocabulary **K**eypoint **D**etection (OVKD) task. It aims to use text prompts to localize arbitrary keypoints of any species. To accomplish this objective, we propose Open-Vocabulary **K**eypoint **D**etection with **S**emantic-feature **M**atching (KDSM), which utilizes both vision and language models to harness the relationship between text and vision and thus achieve keypoint detection through associating text prompt with relevant keypoint features. Additionally, KDSM integrates domain distribution matrix matching and some special designs to reinforce the relationship between language and vision, thereby improving the model's generalizability and performance. Extensive experiments show that our proposed components bring significant performance improvements, and our overall method achieves impressive results in OVKD. Remarkably, our method outperforms the state-of-the-art few-shot keypoint detection methods using a zero-shot fashion. We will make the source code publicly accessible.

## 1 INTRODUCTION

Keypoint detection is a fundamental computer vision task for localizing specific points of interest in images. Although existing animal body and face keypoint detection methods (Khan et al., 2020; Xia et al., 2022; Ma et al., 2022; Xu et al., 2022b; Zhang et al., 2023a) have demonstrated remarkable performance, they are limited to certain species and keypoint categories, necessitating new annotations for previously unseen ones. To address the diversity of the natural world, it is crucial to recognize novel species and keypoint categories. However, traditional approaches depend on time-consuming manual annotation. Zero-shot learning (Xu et al., 2021) has emerged as a valuable technique for enabling recognition without prior annotation, particularly for rare or less-studied species and keypoint categories. Alternative strategies, such as class-agnostic keypoint detection methods (Snell et al., 2017; Nakamura & Harada, 2019; Xu et al., 2022a) (see Fig. 1), aim to detect arbitrary keypoints using few-shot learning based on a small number of annotated images. Despite their versatility, these methods lack the zero-shot capability to recognize keypoints without any prior annotation.

To address these limitations, we introduce a novel task called **O**pen-**V**ocabulary **K**eypoint **D**etection (OVKD), which aims to overcome the constraints of existing keypoint detection methods. OVKD framework aims to detect arbitrary ({*animal species*}, {*keypoint category*}) pairs, including those that are absent from the training data. The recent advancements in vision-language models such as (Radford et al., 2021; Jia et al., 2021) have demonstrated the ability to capture relationships between textual and visual data, thus allowing these models to comprehend contextual information from textual prompts. Inspired by that, we propose language-driven OVKD (unless specified otherwise, OVKD refers to language-driven OVKD), which detects arbitrary keypoints in arbitrary animal species using textual prompts guidance.

The most straightforward approach to addressing the language-driven OVKD is to employ a simple baseline framework that utilizes language models for obtaining robust textual embeddings and performing comprehensive keypoint detection. Specifically, this approach utilizes textual embeddings to transform visual features (i.e., multiplied by textual embeddings) to generate keypoint heatmaps. However, OVKD requires not only a global understanding of the image but also the ability to locate specific keypoints, which the baseline framework may not accomplish. Moreover, the baseline framework's generalization capability across various species and keypoint categories is inadequate. This occurs because the method fails to fully capture the intricate relationships between text and visual features, among different keypoints, and across various species, which are crucial for OVKD.

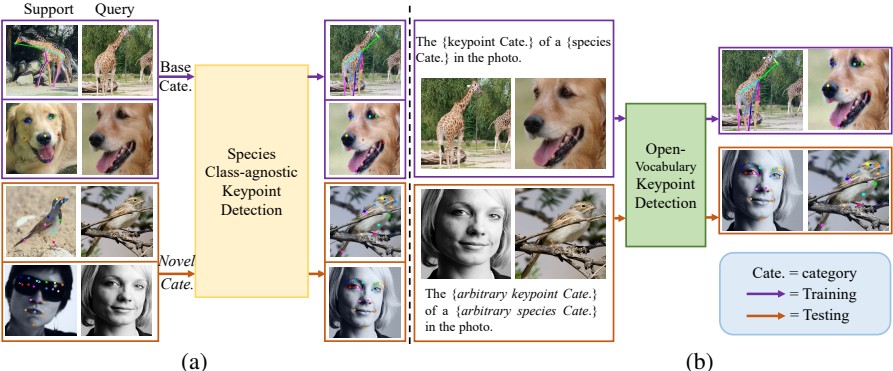

Figure 1: Species Class-Agnostic Keypoint Detection vs. Language-driven Open-Vocabulary Keypoint Detection. (a) Current class-agnostic keypoint detection needs support images for guidance during training and testing to detect keypoints in new species. (b) Language-driven OVKD aims to use text prompts that embed both $\{animal\ species\}$ and $\{keypoint\ category\}$ as semantic guidance to localize arbitrary keypoints of any species.

To address the limitations of the baseline framework, we present a novel framework called Open-Vocabulary **K**eypoint **D**etection with **S**emantic-feature **M**atching (KDSM). KDSM overcomes the challenges of OVKD and the drawbacks of baseline by leveraging domain distribution matrix matching to transform keypoint detection into a problem of aligning textual semantic feature distributions from input text prompts with the detected keypoints' heatmaps. Specifically, domain distribution matrix matching can map the relationship between textual features and detected keypoint heatmaps into a shared semantic space, thus effectively capturing their relationships and improving the generalization ability of KDSM. Moreover, in contrast to simple multiplication in the baseline, our matching can exploit the relationship between different keypoints and species during training, leading to better generalizability during testing for unseen keypoints and species. We adopt a vision-keypoint relational awareness module that calibrates visual features based on their semantic relationships with keypoints. This module comprises self-attention and cross-attention mechanisms, adept at capturing connections between visual and textual information and facilitating the learning process.

We conduct extensive experiments to evaluate the effectiveness of our proposed method. The results indicate that KDSM attains remarkable performance in OVKD, substantially surpassing the baseline framework. Moreover, our method is zero-shot and even surpasses the state-of-the-art few-shot keypoint detection method. In summary, our main contributions are as follows:

- We present the task of OVKD, which aims to use text prompts to detect arbitrary keypoint categories of any animal species without the need for labeled data.

- We present KDSM, an effective OVKD framework that leverages the benefit of powerful language models to exploit the relationship between text and vision. Moreover, KDSM employs domain distribution matrix matching and some special designs to improve the OVKD performance significantly.

- Extensive experimental results demonstrate the effectiveness of KDSM, achieving remarkable performance in OVKD and outperforming the baseline framework by a large margin. Furthermore, KDSM even surpasses the state-of-the-art few-shot keypoint detection method.

## 2 RELATED WORKS

### 2.1 CLASS-AGNOSTIC KEYPOINT DETECTION

With the development of keypoint detection (Newell et al., 2016; Wang et al., 2020; Zhang et al., 2023a; Xu et al., 2022b; Geng et al., 2023), recent research (Xu et al., 2022a) developed class-agnostic keypoint detection techniques that can identify keypoints in various animal species without category-specific training. However, these techniques typically rely on "support images" as guidance during both training and testing phases, similar to FS-ULUS (Lu & Koniusz, 2022), ProtoNet (Snell et al., 2017), MAML (Finn et al., 2017), and Fine-tune (Nakamura & Harada, 2019). This reliance limits their effectiveness when encountering previously unseen species or keypoints. CLAMP (Zhang et al., 2023b) introduces prompt-based contrastive learning for animal pose estimation, however,

they limited the language space to different animal species but with the same keypoint categories. The method and experimental design of CLAMP are difficult to adapt to large species and keypoint categories changes like method (Xu et al., 2022a; Lu & Koniusz, 2022), let alone achieve universal animal body and facial keypoint detection like us. Different with these methods, we propose the **O**pen-**V**ocabulary **K**eypoint **D**etection (OVKD) task, which eliminates the need for support images by utilizing text prompts that include the $\{animal\ species\}$ and $\{keypoint\ category\}$ for semantic guidance. It aims to detect arbitrary keypoints of any species, following the zero-shot learning principle.

## 2.2 OPEN-VOCABULARY LEARNING

Open-vocabulary learning has been explored in several computer vision tasks, including object detection (Zareian et al., 2021; Bangalath et al., 2022; Yao et al., 2022), semantic segmentation (Xu et al., 2021; Li et al., 2022; Ghiasi et al., 2022), video classification (Ni et al., 2022; Qian et al., 2022), and so on. Recent advances in vision-language models such as CLIP (Radford et al., 2021) and ALIGN (Jia et al., 2021) have demonstrated their potential in tasks that require a joint understanding of visual and textual information conducive to open-world learning. While existing open-vocabulary researches excel in image-level classification (Zareian et al., 2021), per-pixel classification (Li et al., 2022), and mask classification (Xu et al., 2021) tasks, keypoint detection presents a greater challenge, requiring global image understanding and local localization. To address this challenge, we propose "Domain Distribution Matrix Matching" to transform the keypoint detection into a problem of matching semantic feature distribution from input text prompts with detected heatmaps.

## 3 METHOD

### 3.1 PROBLEM FORMULATION: OPEN-VOCABULARY KEYPOINT DETECTION (OVKD)

The **O**pen-**V**ocabulary **K**eypoint **D**etection (OVKD) task for animal (including human) body and face keypoint localization is a generalized zero-shot task involving arbitrary keypoint category and arbitrary animal species. The OVKD task aims to train a framework to detect keypoints in images, even if the animal species or keypoint category does not appear in training data. With the advancements in vision-language models such as CLIP (Radford et al., 2021), we introduce to take advantage of powerful language models to achieve OVKD, i.e., language-driven OVKD (unless specified otherwise, OVKD refers to language-driven OVKD).

For the language-driven OVKD, textual prompts are leveraged to guide the framework in understanding the semantic information and locating specific keypoints. Assuming we have a training set $\mathcal{D}_{train}$ and a test set $\mathcal{D}_{test}$, $\mathcal{D}_{train} = \{(\mathbf{I}, P(s_i, k_j))\}_{i=1,j=1}^{N_s, N_k}$, $\mathcal{D}_{test} = \{(\mathbf{I}, P(s_i', k_j'))\}_{i=1,j=1}^{N_s', N_k'}$. Here, $\mathbf{I}$ represents images, and $P(s_i, k_j)$ denotes the text prompts constructed based on species $s_i$ and keypoint category $k_j$. $N_s$ and $N_k$ represent the number of species and keypoint categories in the training set, respectively, while $N_s'$ and $N_k'$ represent the number of species and keypoint categories in the test set, respectively. The test set comprises unseen animal species and keypoint categories, which requires the detector to estimate arbitrary keypoints according to the textual prompts.

### 3.2 BASELINE: A SIMPLE FRAMEWORK FOR OVKD

In order to tackle the challenging **O**pen-**V**ocabulary **K**eypoint **D**etection (OVKD) task, we build a baseline framework that is able to predict arbitrary keypoint category of any animal species in a flexible way as shown in Fig. 2. The baseline method constructs text prompts for the OVKD task and extracts textual embedding using a Text_Encoder. The Vision_Encoder is applied to extract visual features of the input image simultaneously. Then, the visual and textual features are integrated together to output heatmaps of required keypoints.

**Text Prompts Construction.** In this step, we construct text prompts by template "The $\{keypoint\ category\}$ of a $\{animal\ species\}$ in the photo." to help language models understand the proposed task more effectively. For instance, with "giraffe" as the animal species and "neck" as the keypoint category, the prompt would be: "The $\{neck\}$ of a $\{giraffe\}$ in the photo.". The same template is used for different animals and keypoints, replacing placeholders accordingly. This template can help the language model to focus on the relationship between animal species and keypoints. It also enables easy generalization to new animals and keypoints in the open-vocabulary settings.

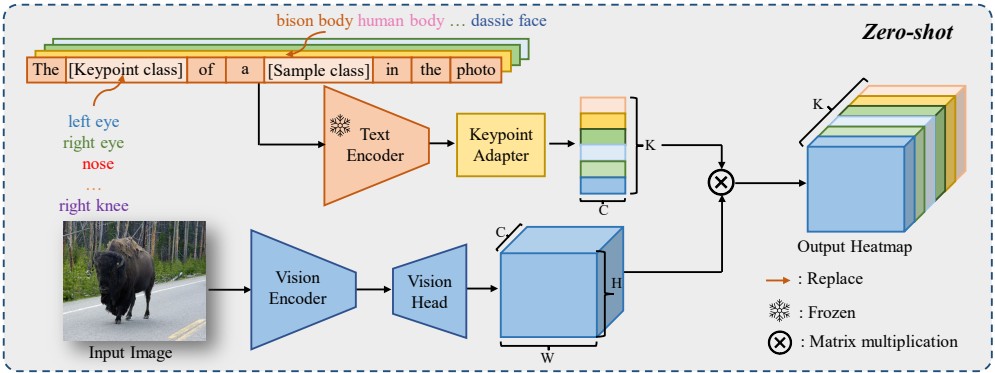

Figure 2: An overview of the baseline method for OVKD. The baseline comprises a Vision_Encoder and Text_Encoder, and a Keypoint_Adapter.

**Text Feature Extraction.** Employing the pre-trained CLIP text encoder (Radford et al., 2021), we process the preprocessed text prompts $T = (T_1, T_2, ..., T_K)$ for an image with $K$ target keypoint categories:

$$\mathbf{T} = \text{Keypoint\_Adapter}(\text{Text\_Encoder}(T)), \tag{1}$$

where $\text{Text\_Encoder}(T) \in \mathbb{R}^{K \times C_0}$ represents the extracted textual features. $\text{Keypoint\_Adapter}$ is a two-layer Multilayer Perceptron (MLP) used to refine these features and make them compatible with the image feature representations, resulting in a refined semantic feature space $\mathbf{T} \in \mathbb{R}^{K \times C}$ ($K = 100, C = 64$ in our implementation), because different species have different numbers of key points, assuming there are actually $k$ key points, the $K - k$ text prompts without keypoint meaning is set to "There is not the key-point we are looking for.".

**Vision Feature Extraction.** Given an input image $I$, we train a Vision_Encoder and a Vision_Head to extract image features:

$$\mathbf{V} = \text{Vision\_Head}(\text{Vision\_Encoder}(I)), \tag{2}$$

where $\mathbf{V} \in \mathbb{R}^{C \times H \times W}$ ($H = 64, W = 64$ in our implementation) represents vision feature. We utilize pre-trained models, such as ResNet (He et al., 2016) and ViT (Dosovitskiy et al., 2020), as the backbone of the Vision_Encoder. These models are known to be effective in extracting hierarchical visual features from images. The Vision_Head, inspired by SimpleBaseline (Xiao et al., 2018), is composed of three deconvolutional layers. These layers serve to upsample the low-resolution feature maps acquired from the image encoder, thereby successfully recovering spatial information and enabling accurate keypoint localization.

**Keypoint Heatmap Prediction.** The objective of OVKD is to predict keypoint localization by aligning semantic textual and spatial visual features. In order to calculate the similarity between the textual concept and pixel-level visual representation, the extracted features are combined through matrix multiplication:

$$\mathbf{H} = \mathbf{T} \times \mathbf{V}, \tag{3}$$

where $\mathbf{H} \in \mathbb{R}^{K \times H \times W}$ denotes predicted heatmaps. The model is supervised by Mean Squared Error (MSE) loss between the predicted heatmaps $\mathbf{H}$ and the ground truth heatmaps $\mathbf{G} \in \mathbb{R}^{K \times H \times W}$. During the training process, the weights of the Text_Encoder remain fixed, while all other parameters are updated. The refined semantic features $\mathbf{T}$ act as the prototype when combined with the vision features $\mathbf{V}$. The matrix multiplication operation conducts a transformation of visual features to the output heatmap spaces, driven by the semantic information contained in the text prompts.

### 3.3 OPEN-VOCABULARY KEYPOINT DETECTION WITH SEMANTIC-FEATURE MATCHING

In this section, we propose a novel framework, namely Open-Vocabulary **K**eypoint **D**etection with **S**emantic-feature **M**atching (KDSM), to address the limitations of the baseline OVKD framework. The baseline framework struggles with generalization across species and keypoint categories, and fails to either capture the complex relationship between textual and visual features or establish the connections between the animal species, leading to sub-optimal keypoint detection performance. Therefore, KDSM proposes domain distribution matrix matching and adopts a vision-keypoint relational awareness module to address the above problems. As shown in Fig. 3, KDSM first

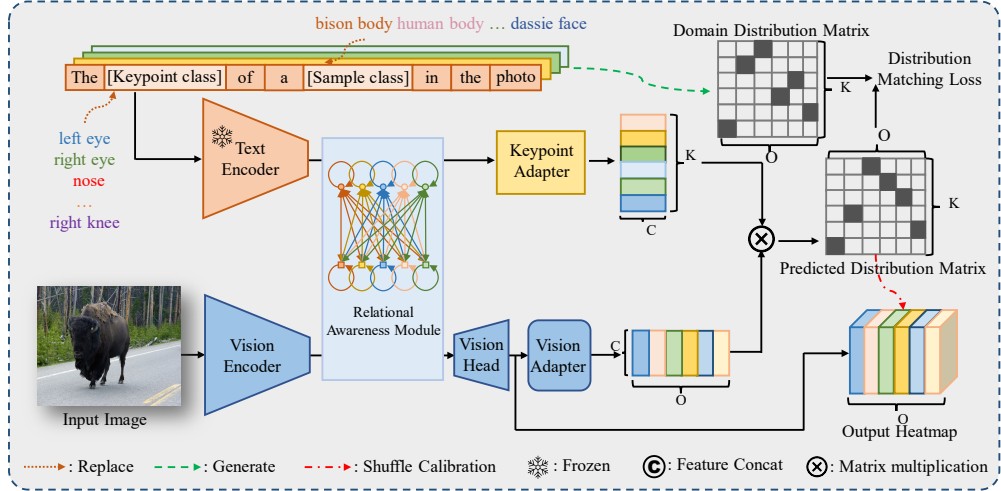

Figure 3: An overview of KDSM. KDSM comprises a vision_encoder, a text_encoder, a keypoint_adapter, and a vision_head similar to the baseline. Relational awareness module adjusts visual features according to their associations with keypoints.

constructs the text prompts and extracts text features in a similar way as the baseline. Then, the vision-keypoint relational awareness module is applied to perform interaction between visual features and textual embeddings for vision-text representation alignment. Finally, domain distribution matrix matching is proposed to capture cross-species keypoint-level relationships to further enhance the generalization ability of KDSM. In particular, KDSM supports multiple text prompt inputs to detect multiple keypoints at each time.

**Vision-keypoint Relational Awareness Module.** In the proposed vision-keypoint relational awareness module, we utilize a series of Transformer blocks to effectively capture the relationships between visual and textual information inspired by (Pan et al., 2020). It consists of two major components: Self-Attention (Vaswani et al., 2017) (Self_Attn.) and Cross-Attention (Carion et al., 2020) (Cross_Attn.). The self-attention layers aim to facilitate the interaction between text embeddings of the input sample. The self-attention layers aggregate the keypoint features as follows:

$$\mathbf{Y}_t = \text{Self\_Attn.}(\text{Text\_Encoder}(\mathbf{T})). \tag{4}$$

The refined keypoint features $\mathbf{Y}_t$ capture the relationship of textual semantic concepts between the keypoints of a specific species, which is aligned with visual features containing rich context and suitable for visual-keypoint interaction.

For the cross-attention layers, the query is the output features from the Vision_Encoder, while the key and value are the refined features $\mathbf{Y}_t$. The cross-attention layers interacts the context-aware visual features Vision_Encoder($I$) with the refined features $\mathbf{Y}_t$ to update vision representation:

$$\bar{\mathbf{V}} = \text{Cross\_Attn.}(\text{Vision\_Encoder}(I), \mathbf{Y}_t) \tag{5}$$

The updated visual features $\bar{\mathbf{V}}$ effectively capture the relationships between visual and keypoint information, bridging the gap between keypoint and vision representation.

**Domain Distribution Matrix Matching.** There exists cross-species commonality at the keypoint category level for OVKD since the keypoints of different animals may be similar. The similarity could be grasped during training by dividing all the keypoint categories into several groups and learning keypoint categories in the same group together. During testing, the common characteristics of a certain group could be transferred to an unseen keypoint category by setting it into an appropriate group. To this end, domain distribution matrix matching benefits the prediction of unseen keypoint categories of arbitrary species by borrowing knowledge from seen species and keypoints.

In order to capture the cross-species keypoint-level relationships, we propose a domain distribution matrix that establishes the connection between the keypoint categories and output heatmaps. With the domain distribution matrix, each keypoint category is divided into a certain group, and each group is trained on a fixed channel of heatmaps. We perform K-means clustering on all the keypoint categories in the training set and cluster them into $O$ groups according to their textual embeddings. The textual embeddings are obtained by applying pre-trained language model to the words $\{keypoint\ category\}$.

For each training sample, we pre-compute its *binary* domain distribution matrix $\mathbf{D} \in \mathbb{R}^{K \times O}(O = 100$ in our implementation) according to the keypoint categories of that sample, where $K$ is a fixed number that is no smaller than the maximum number of keypoints of any sample. $\mathbf{D}_{ij} = 1$ only if the $i$-th keypoint belongs to the $j$-th group. If the number of keypoints $K'$ of the sample is smaller than $K$, we set $\mathbf{D}_{ij} = 0$ for $i \in [K' + 1, K], j \in [1, O]$.

We predict the distribution matrix to learn the ability for group selection. First, the updated visual features $\bar{\mathbf{V}}$ are concatenated with original visual features to enhance the visual representation. The concatenated features are processed by the $\text{Vision\_Head}$ to obtain the heatmaps $\mathbf{H}' \in \mathbb{R}^{O \times H \times W}$. Then, the $\text{Vision\_Adapter}$ is applied to obtain the adjusted visual features $\mathbf{V}'$ from the heatmaps, and the $\text{Keypoint\_Adapter}$ is applied for adjusted textual embeddings $\mathbf{T}'$ from the original textual embeddings. Finally, we calculate the similarity between adjusted visual features $\mathbf{V}' \in \mathbb{R}^{C \times O}$ and adjusted textual embeddings $\mathbf{T}' \in \mathbb{R}^{K \times C}$ to form a predicted distribution matrix $\mathbf{P} \in \mathbb{R}^{K \times O}$:

$$\mathbf{P} = \mathbf{T}' \times \mathbf{V}'. \tag{6}$$

**Loss Function.** The matching loss $L_{match}$ is calculated as the cross-entropy loss between the predicted distribution matrix $\mathbf{P}$ and the domain distribution matrix $\mathbf{D}$ to learn the alignment between keypoint categories and heatmap channels:

$$L_{match} = -\sum_{i=1}^{K} \sum_{j=1}^{O} \mathbf{D}_{ij} \log \mathbf{P}_{ij}. \tag{7}$$

In order to learn the localization ability of KDSM, the heatmaps $\mathbf{H}'$ generated by $\text{Vision\_Head}$ are reordered among channels according to the domain distribution matrix. To reorder $\mathbf{H}'$, we use the domain distribution matrix $\mathbf{D}$. The index of the element 1 in $i^{th}$ row of $\mathbf{D}$ is $o$, which indicates that the $o^{th}$ channel in $\mathbf{H}'$ is the $i^{th}$ channel in $\mathbf{H}$. Pytorch has many functions for record indexing and reordering, such as "torch.index_select". The reordered heatmaps $\mathbf{H} \in \mathbb{R}^{O \times H \times W}$ is supervised by the ground truth heatmaps $\mathbf{G} \in \mathbb{R}^{O \times H \times W}$ via Mean Squared Error (MSE) loss. The first $k$ channels of $\mathbf{G}$ correspond to the heatmaps generated by the corresponding keypoint positions of the $k$ input text prompts. We simply set the other channels of the heatmaps behind it are invalid 0 matrices. The total loss function to train KDSM is defined as follows:

$$L_{total} = \alpha L_{match} + \beta MSE(\mathbf{H}, \mathbf{G}), \tag{8}$$

where $\alpha$ and $\beta$ are the balance weights, and they are set to 0.001 and 1.0 unless otherwise specified. We follow (Xu et al., 2022a) to built $\mathbf{G}$, and the size of $\mathbf{G}$ is the same as $\mathbf{H} \in \mathbb{R}^{O \times H \times W}$ ($O = 100, H = 64, W = 64$ in our implementation). Only the first $k$ heatmaps ($k$ is the number of valid text prompts) are valid for $\mathbf{G}$, and the $i^{th}$ channel of $\mathbf{G}$ corresponds to the $i^{th}$ prompt. The other $(O - k)$ heatmaps are set to zero.

**Inference Process.** During inference, given an input image and text prompts, KDSM follows the training process to estimate the heatmaps and predicted distribution matrix. For each keypoint category $k$, we search the maximum value in the $k$-th row of the predicted distribution matrix as the index of the corresponding heatmap channel. The reordered heatmaps are calibrated according to the indexes and are used as the prediction results. The keypoint localization is decoded as the coordinates with the maximum scores in the reordered heatmaps.

## 4 EXPERIMENTS

### 4.1 OPEN-VOCABULARY EVALUATION PROTOCOL

**Dataset Split.** MP100 (Xu et al., 2022a) is introduced for category-agnostic pose estimation, which contains over 20K instances covering 100 sub-categories and 8 super-categories (human hand, human face, animal body, animal face, clothes, furniture, and vehicle). However, some of the keypoint categories in MP100, such as those for clothes and furniture, lack practical semantic information and are not suitable for language-driven OVKD. Thus, we selected a subset of 78 animal categories (including human) with keypoint annotations that have specific, meaningful semantic information. We call this subset "MP-78", including COCO (Lin et al., 2014), AFLW (Koestinger et al., 2011), OneHand10K (Wang et al., 2018), AP-10K (Yu et al., 2021), Desert Locust (Graving et al., 2019), MascaquePose (Labuguen et al., 2021), Vinegar Fly (Pereira et al., 2019), AnimalWeb (Khan et al.,

Table 1: Comparisons with the baseline framework on the MP-78 dataset for "Diverse Keypoint Categories" setting. KDSM significantly outperforms the baseline.

| Framework | Split1 | Split2 | Split3 | Split4 | Split5 | Mean(PCK) |
|-----------|--------|--------|--------|--------|--------|-----------|
| Baseline  | 42.02  | 44.00  | 42.55  | 43.80  | 42.26  | 42.93     |
| KDSM      | **79.02** | **71.35** | **76.68** | **79.79** | **74.67** | **76.30** |

Table 2: Comparisons on MP-78 dataset for "Varied Animal Species" setting. Notably, KDSM outperforms all other methods, including few-shot approaches.

| Framework | Shot setting | Split1 | Split2 | Split3 | Split4 | Split5 | Mean(PCK) |
|-----------|--------------|--------|--------|--------|--------|--------|-----------|
| MAML (Finn et al., 2017) | 1-shot | 75.11 | 74.31 | 69.80 | 68.22 | 67.44 | 70.98 |
| Fine-tune (Nakamura & Harada, 2019) | 1-shot | 76.65 | 76.41 | 71.37 | 69.97 | 69.36 | 72.75 |
| FS-ULUS (Lu & Koniusz, 2022) | 1-shot | 73.69 | 70.65 | 63.97 | 71.14 | 63.65 | 68.62 |
| POMNet (Xu et al., 2022a) | 1-shot | 73.07 | 77.89 | 71.79 | 78.76 | 70.26 | 74.35 |
| MAML (Finn et al., 2017) | 5-shot | 76.37 | 75.53 | 71.15 | 69.46 | 67.55 | 72.01 |
| Fine-tune (Nakamura & Harada, 2019) | 5-shot | 77.81 | 76.51 | 72.55 | 71.09 | 69.85 | 73.56 |
| FS-ULUS (Lu & Koniusz, 2022) | 5-shot | 78.34 | 79.67 | 76.89 | 81.52 | 75.23 | 78.33 |
| POMNet (Xu et al., 2022a) | 5-shot | 81.25 | 86.44 | 81.01 | **86.93** | 78.68 | 82.86 |
| Baseline | zero-shot | 56.06 | 55.36 | 54.35 | 53.07 | 50.66 | 53.90 |
| KDSM | zero-shot | **84.02** | **87.99** | **83.22** | 83.20 | **80.25** | **83.74** |

2020), CUB-200 (Welinder et al., 2010). Finally, MP-78 contains over 14K images and 15K annotations[1].

In order to evaluate the generalization ability of OVKD to different keypoint categories and animal species, we design two settings, that is "Diverse Keypoint Categories" for new $\{keypoint\ category\}$, and "Varied Animal Species" for new $\{animal\ species\}$ like (Xu et al., 2022a). All zero-shot settings fall under "transductive generalized zero-shot learning (Pourpanah et al., 2022)". The specific zero-shot experimental settings are in the appendix materials.

**Evaluation Metrics.** In our study, we use a widely accepted metric for quantifying keypoint detection accuracy, i.e., the Probability of Correct Keypoint (PCK). To determine whether a predicted keypoint is correct or not, we compare the normalized distance between the estimated keypoint location and the actual ground-truth keypoint location against a predefined threshold ($\sigma$). In our experiments, we report the mean PCK@0.2 (with $\sigma$ set to 0.2) for all categories in each individual split. The definition of PCK is the same as POMNet (Xu et al., 2022a). Furthermore, to reduce the potential impact of category bias, we include the average PCK result across all splits.

## 4.2 IMPLEMENTATION DETAILS

The objects of interest are extracted using their bounding boxes and resized to dimensions of $256 \times 256$. To bolster the model's generalization capabilities, data augmentation techniques such as random scaling (varying from $-15\%$ to $15\%$) and random rotation (varying from $-15°$ to $15°$) are applied. Training is carried out across 4 GPUs, each with a batch size of 64, for a total of 210 epochs. The Adam optimizer (Kingma & Ba, 2014) is employed, starting with an initial learning rate of 1e-3. This learning rate is subsequently decreased to 1e-4 at the $170^{th}$ epoch and to 1e-5 at the $200^{th}$ epoch.

## 4.3 RESULTS FOR OVKD

**Setting A: Diverse Keypoint Categories.** Table 1 presents the performance comparison between the baseline framework and KDSM on the MP-78 dataset for this setting. As shown in the table, KDSM consistently outperforms the baseline across all five splits. The quantitative comparison of the results shows a significant performance improvement when using the KDSM framework. The mean PCK score across all five splits increases from 42.93% for the baseline to 76.30% for the KDSM framework, which corresponds to a 33.37 percentage point enhancement. This indicates that the KDSM approach is more effective at handling the "Diverse Keypoint Categories" setting in the zero-shot setting. The superior performance of the KDSM framework on the "Diverse Keypoint Categories" setting can be attributed to its capacity to better align and match semantic information from textual prompts with visual features, as well as its ability to effectively transfer knowledge from seen keypoint categories to unseen ones.

---

[1]We use the ChatGPT (Brown et al., 2020) to query the names of datasets that do not have semantic naming. The annotations for the two experiment settings of the MP-78 dataset will be released along with our code.

Table 3: Ablation study of proposed components on MP-78 for OVKD. "Diverse Keypoint Categories" is selected as the experiment setting. VKLA and DDMM represent "Vision-keypoint Relational Awareness module" and "Domain Distribution Matrix Matching", respectively.

| Baseline | DDMM | VKLA | Split1 | Split2 | Split3 | Split4 | Split5 | Mean(PCK) |
|---|---|---|---|---|---|---|---|---|
| ✔ | ✘ | ✘ | 42.02 | 44.00 | 42.55 | 43.80 | 42.26 | 42.93 |
| ✔ | ✔ | ✘ | 69.64 | 57.86 | 67.95 | 62.10 | 71.92 | 65.89 |
| ✔ | ✔ | ✔ | **79.02** | **71.35** | **76.68** | **79.79** | **74.67** | **76.30** |

Table 4: Performance comparison of different attention blocks in Setting "Diverse Keypoint Categories" for the OVKD task.

| Self_Attn. | Cross_Attn. | Split1 | Split2 | Split3 | Split4 | Split5 | Mean(PCK) |
|---|---|---|---|---|---|---|---|
| 1 | 3 | 65.43 | 53.78 | 48.80 | 56.90 | 57.97 | 56.58 |
| 2 | 3 | 74.89 | 61.87 | 69.55 | 78.56 | 70.39 | 71.05 |
| 3 | 3 | 79.02 | 71.35 | 76.68 | 79.79 | 74.67 | 76.30 |
| 4 | 3 | 82.49 | 83.15 | 72.00 | 76.66 | 74.33 | 77.73 |
| 3 | 1 | 77.04 | 71.16 | 65.19 | 69.18 | 66.65 | 69.84 |
| 3 | 2 | 79.44 | 69.07 | 78.38 | 76.49 | 73.75 | 74.43 |
| 3 | 4 | 79.62 | 67.00 | 75.69 | 76.41 | 71.71 | 74.09 |

**Setting B: Varied Animal Species.** Table 2 displays the performance comparison between the baseline framework and KDSM on the MP-78 dataset for the "Varied Animal Species" setting under a zero-shot setting. Additionally, it compares the results with class-agnostic keypoint detection methods under 1-shot and 5-shot settings.

The KDSM framework significantly outperforms the baseline in the zero-shot setting, demonstrating its effectiveness in handling unseen animal species without category-specific training. Recent research (Nakamura & Harada, 2019; Xu et al., 2022a) has developed class-agnostic keypoint detection techniques, like POMNet (Xu et al., 2022a), that can identify keypoints across various animal species without category-specific training. However, these techniques typically rely on support images during both training and testing phases, which limits their effectiveness when encountering previously unseen species or keypoints. In contrast, our OVKD approach using the KDSM framework removes the need for support images by leveraging text prompts that include the $\{animal\ species\}$ and $\{keypoint\ category\}$ for semantic guidance. This enables our approach to detect unseen species or keypoints for a wide range of animal species in a zero-shot learning fashion. When comparing to the 1-shot and 5-shot settings of Fine-tune and POMNet, KDSM demonstrates superior performance. Specifically, KDSM surpasses POMNet's 1-shot setting with 9.39% and outperforms POMNet's 5-shot setting with a slight increase of 0.88%. These results indicate that KDSM is effective at detecting keypoints in unseen $\{animal\ species\}$ even without relying on support images, against class-agnostic keypoint detection methods.

The enhanced performance of the KDSM framework can be attributed to its ability to generalize better to unseen categories within seen species due to the efficient knowledge transfer from seen to unseen $\{animal\ species\}$. Overall, KDSM exhibits strong performance in the OVKD for animal species compared to both the baseline and other class-agnostic keypoint detection methods.

## 4.4 ABLATION STUDY

**Domain Distribution Matrix Matching (DDMM).** As shown in Table 3, DDMM significantly improves mean PCK scores from 42.93% to 65.89%, which demonstrates the effectiveness of the DDMM in facilitating knowledge transfer between seen and unseen keypoint categories. Additionally, the consistent performance gains across all splits indicate that the proposed approach is robust and generalizable to various scenarios, further highlighting its potential for real-world applications.

**Vision-keypoint relational Awareness (VKLA) Module.** In Table 3, we observe that when the baseline framework is combined with both the DDMM and VKLA components, there is a significant improvement in mean PCK scores. Specifically, the mean PCK score increases from 42.93% for the baseline framework without these components, to 76.30% with both the DDMM and VKLA modules included. This highlights the importance of the VKLA module in our approach, as it effectively captures the semantic relationship between visual features and textual prompts, leading to better generalization across unseen keypoint categories.

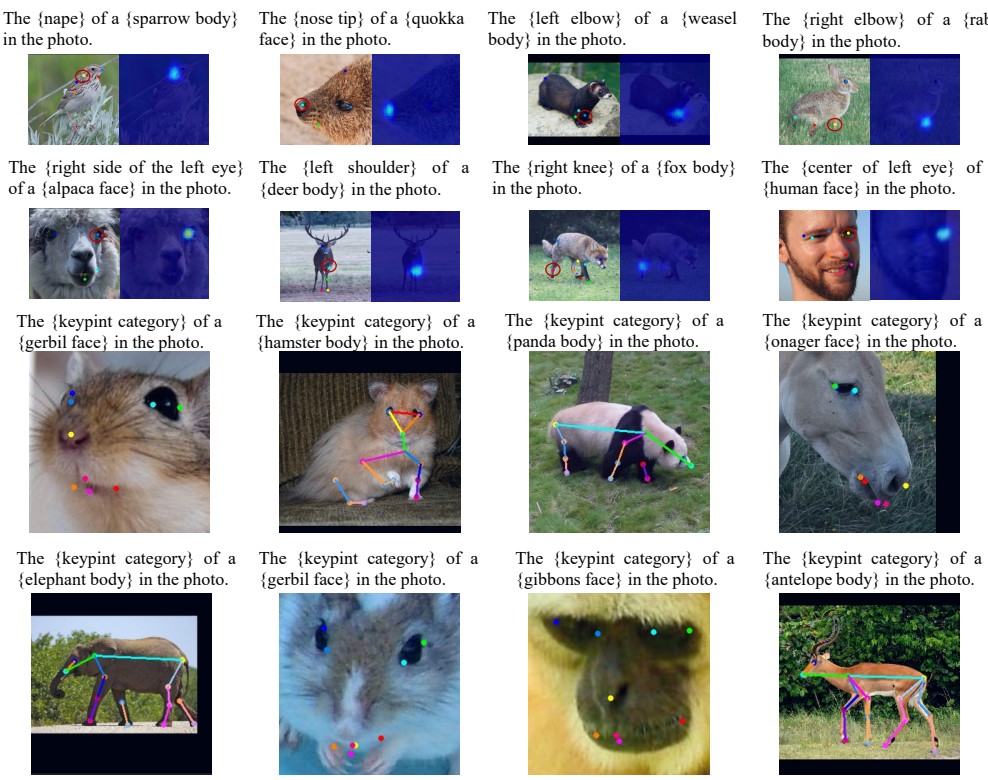

The {nape} of a {sparrow body} in the photo.

The {nose tip} of a {quokka face} in the photo.

The {left elbow} of a {weasel body} in the photo.

The {right elbow} of a {rabbit body} in the photo.

The {right side of the left eye} of a {alpaca face} in the photo.

The {left shoulder} of a {deer body} in the photo.

The {right knee} of a {fox body} in the photo.

The {center of left eye} of a {human face} in the photo.

The {keypint category} of a {gerbil face} in the photo.

The {keypint category} of a {hamster body} in the photo.

The {keypint category} of a {panda body} in the photo.

The {keypint category} of a {onager face} in the photo.

The {keypint category} of a {elephant body} in the photo.

The {keypint category} of a {gerbil face} in the photo.

The {keypint category} of a {gibbons face} in the photo.

The {keypint category} of a {antelope body} in the photo.

Figure 4: Visual results of KDSM on the test sets of two experiment settings of OVKD. The first row shows the heatmaps for new keypoint categories, and the second row shows the results for new species categories. KDSM achieve satisfactory results in both two settings.

We also research the number of self-attention layers and cross-attention layers in the VKLA module. The results are shown in Table 4. We observed that increasing the number of self-attention blocks from 1 to 3 resulted in a noticeable improvement in performance (row 1 vs. row 3). However, further increasing the number of self-attention blocks to 4 did not yield significant gains (row 3 vs. row 4). For the number of cross-attention blocks, we have a similar observation and choose to use three cross-attention blocks in our implementation.

## 4.5 QUALITATIVE RESULTS

In Fig. 4, we show the results of KDSM on two experiment settings of OVKD. The first row shows the heatmaps for new keypoint categories, and the second row shows the keypoint detection results. It can be observed that KDSM is capable of completing the OVKD task in both two settings.

## 5 CONCLUSION

We address the limitations of existing image-based keypoint detection methods for animal (including human) body and face by introducing the task of **O**pen-**V**ocabulary **K**eypoint **D**etection (OVKD). OVKD aims to detect keypoints in images, even if the animal species or keypoint category does not appear in training data. We propose Open-Vocabulary **K**eypoint **D**etection with **S**emantic-feature **M**atching (KDSM), a novel and effective OVKD framework that takes advantage of powerful vision-language models to harness the relationship between text and vision. In particular, KDSM incorporates domain distribution matrix matching and some special designs to improve performance significantly, with a 33.37-point improvement for diverse keypoint categories, and a 29.84-point improvement for varied animal species. Remarkably, KDSM outperforms the state-of-the-art few-shot keypoint detection methods using a zero-shot fashion. The proposed approach lays the groundwork for future exploration and advancements in OVKD, driving further improvements in quantitative performance metrics.

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

# A APPENDIX

## A.1 FURTHER EXPLANATION OF ANIMAL SPECIES DEFINITIONS

In our experimental section, we discussed a subset of MP100 (Xu et al., 2022a) with keypoint annotations that convey specific, meaningful semantic information. This subset is referred to as "MP-78". However, it is important to clarify that in this paper, the term {*animal species*} actually refers to a combination of "target keypoint detection task + animal species." For example, the face and body of a dog are considered two separate {*animal species*} (i.e., "dog face" and "dog body") depending on the target keypoint detection task. In other words, rather than solely referring to distinct animal species, we define species in the context of "keypoint detection task"-specific categories within those animals. This distinction allows us to better focus on the challenges associated with each target task when evaluating the performance of KDSM and other few-shot methods.

Table 5: Performance comparison of Text_Encoder trained with different Vision_Encoder configurations in Setting A for the OVKD task.

| Text_Encoder | Vision_Encoder | Split1 | Split2 | Split3 | Split4 | Split5 | Mean(PCK) |
|---|---|---|---|---|---|---|---|
| Paired with ResNet50 | ResNet50 | 60.60 | 50.34 | 65.61 | 60.07 | 40.89 | 55.50 |
| Paired with ViT-B/32 | ResNet50 | 79.02 | 71.35 | 76.68 | 79.79 | 74.67 | 76.30 |
| Paired with ViT-B/16 | ResNet50 | **82.39** | **72.58** | **73.69** | **80.89** | **81.82** | **78.27** |

Table 6: Impact of hyperparameter settings on the performance of KDSM in Setting A for the OVKD task.

| $\alpha$ | $\beta$ | Split1 | Split2 | Split3 | Split4 | Split5 | Mean(PCK) |
|---|---|---|---|---|---|---|---|
| 1 | 1 | 13.57 | 13.22 | 13.26 | 12.80 | 13.78 | 13.32 |
| $1\times10^{-1}$ | 1 | 42.87 | 31.06 | 32.34 | 14.16 | 31.32 | 30.35 |
| $1\times10^{-3}$ | 1 | 79.02 | 71.35 | 76.68 | 79.79 | 74.67 | 76.30 |
| $1\times10^{-4}$ | 1 | 83.99 | 79.96 | 87.50 | 87.32 | 85.86 | 84.93 |
| $1\times10^{-6}$ | 1 | 87.93 | 88.50 | 87.64 | 88.28 | 88.82 | **88.23** |
| $1\times10^{-7}$ | 1 | 87.71 | 89.47 | 87.33 | 86.40 | 89.02 | 87.99 |
| $1\times10^{-8}$ | 1 | 30.50 | 30.02 | 30.54 | 30.36 | 28.77 | 30.04 |
| $1\times10^{-10}$ | 1 | 29.28 | 31.38 | 30.61 | 31.48 | 29.69 | 30.49 |
| 0 | 1 | 30.02 | 30.63 | 32.64 | 31.31 | 32.17 | 31.35 |
| $1\times10^{-3}$ | $1\times10^{1}$ | 84.03 | 87.14 | 84.99 | 85.14 | 86.30 | 85.52 |
| $1\times10^{-3}$ | $1\times10^{2}$ | 88.07 | 89.21 | 88.18 | 88.93 | 88.99 | 88.68 |
| $1\times10^{-3}$ | $1\times10^{3}$ | 88.87 | 90.04 | 87.94 | 89.57 | 87.17 | **88.72** |
| $1\times10^{-3}$ | $1\times10^{6}$ | 87.74 | 90.05 | 86.76 | 86.59 | 89.11 | 88.05 |
| $1\times10^{-6}$ | $1\times10^{1}$ | 88.15 | 87.81 | 86.05 | 89.47 | 87.44 | 87.78 |
| $1\times10^{-6}$ | $1\times10^{2}$ | 88.63 | 88.20 | 87.92 | 88.66 | 89.29 | 88.54 |
| $1\times10^{-6}$ | $1\times10^{3}$ | 86.95 | 87.75 | 84.52 | 88.73 | 87.90 | 87.17 |

## A.2 MORE IMPLEMENTATION DETAILS

**Text Encoder.** In this paper, unless otherwise specified, we use CLIP (Radford et al., 2021)'s text encoder, which is pre-trained with ViT-B/32 vision encoder using image-text paired data.

**Vision Encoder.** The Vision_Encoder in this paper is set to ResNet50 (He et al., 2016) pre-trained on the ImageNet dataset by default unless otherwise specified.

**Vision-keypoint Relational Awareness Module.** Self_Attn. contains three layers. Each layer has a multi-head self-attention mechanism and a feed-forward neural network (FFN). The self-attention component has four attention heads and an embedded dimension of 512. The dropout rate for the attention mechanism is set to 0.1. The FFN configuration includes two fully connected layers, an embedded dimension of 512, and feedforward channels of 2048. The activation function used is ReLU, and the dropout rate is set to 0.1.

Cross_Attn. also contains three layers. Each layer comprises a multi-head self-attention mechanism, a multi-head cross-attention mechanism, and a feed-forward neural network (FFN). Both attention mechanisms have four heads, an embedded dimension of 512, and a dropout rate of 0.1. The FFN shares the same configuration as in the encoder, with two fully connected layers, an embedded

Table 7: Performance of KDSM on different super-categories in Setting A for the OVKD task.

| Super-Category | Split1 | Split2 | Split3 | Split4 | Split5 | Mean(PCK) |
|---|---|---|---|---|---|---|
| Face | 85.05 | 77.56 | 83.52 | 87.31 | 76.01 | 81.89 |
| Body | 76.73 | 68.61 | 73.67 | 76.37 | 74.49 | 73.97 |
| Face w/ Body | 79.02 | 71.35 | 76.68 | 79.79 | 74.67 | 76.30 |

The {left side of the left eye} of a {quokka face} in the photo.

The {right front paw} of a {cheetah body} in the photo.

The {right side of lip} of a {onager face} in the photo.

The {right knee} of a {bison body} in the photo.

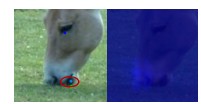

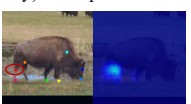

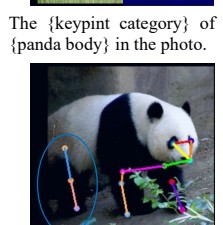

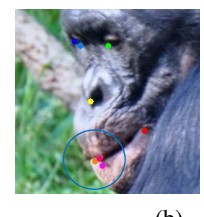

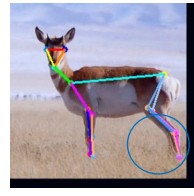

The {keypint category} of a {onager face} in the photo.

The {keypint category} of a {panda body} in the photo.

The {keypint category} of a {bonobo face} in the photo.

The {keypint category} of a {antelope body} in the photo.

(a)        (b)

Figure 5: Visual results of challenging KDSM on the test sets of two experiment settings of OVKD. (a) Demonstrates that KDSM can handle challenging scenarios involving body occlusion, environmental occlusion, and complex poses. (b) Illustrates the failure cases of KDSM in challenging keypoint detection. The points circled in red represent the ground truth keypoint locations corresponding to the heatmaps. The blue circles enclose the challenging regions of keypoint detection.

dimension of 512, and feedforward channels of 2048. The activation function used is ReLU, and the dropout rate is set to 0.1.

## A.3 ZERO-SHOT SETTINGS

In the first experiment setting, "Diverse Keypoint Categories," we divide the keypoint categories associated with each of the 78 species into two parts: seen $\{keypoint\ category\}$ and unseen $\{keypoint\ category\}$. During training, we only used the seen categories, while the unseen categories were reserved for testing. For fair evaluation, we randomly split seen $\{keypoint\ category\}$ for each species to form seen $\{keypoint\ category\}$ sets. We form five different train/test sets splits.

For "Varied Animal Species", MP-78 is split into train/test sets, with 66 $\{animal\ species\}$ for training, and 12 $\{animal\ species\}$ for testing. To ensure the generalization ability of the framework, we evaluate the framework performance on five splits like (Xu et al., 2022a), where each $\{animal\ species\}$ is treated as a novel one on different splits to avoid $\{animal\ species\}$ bias.

## A.4 DISCUSSION ON THE CHOICE OF TEXT ENCODER

In Table 5, we compare the performance of using text encoders pre-trained with different vision encoders on the OVKD task. The results indicate the following average Mean(PCK) scores: ResNet50 with 55.50, ViT-B/32 with 76.30, and ViT-B/16 with 78.27. Among these, the corresponding text encoder of the ViT-B/16 vision encoder achieves the best performance. The performance gap between the text encoders suggests that using a stronger text encoder, i.e., one pre-trained with a stronger vision encoder, yields better results (although in the paper, we used ViT-B/32). Therefore, our method has great potential for improvement by utilizing a stronger text encoder.

## A.5 DISCUSSION OF THE LOSS FUNCTION OF KDSM

In this section, we try more hyperparameter settings. Table 6 shows the impact of hyperparameter settings on the performance of KDSM in the OVKD Setting A evaluation. As $\alpha$ decreases from 1 to $10^{-10}$ while keeping $\beta$ constant at 1, the Mean(PCK) increases, exhibiting an upward trend, with the best performance achieved at $\alpha = 10^{-6}$, yielding a Mean(PCK) of 88.23. However, when $\alpha$ further

decreases beyond $10^{-6}$ or is set to 0, the Mean(PCK) starts to decline, indicating an optimal range for the value of $\alpha$. Especially, when $\alpha$ is set to 0, the Mean(PCK) drops to 31.15, indicating the importance of "Domain Distribution Matrix Matching."

With $\alpha = 10^{-3}$ held constant, increasing $\beta$ to $10^3$ results in a Mean(PCK) of 88.72. However, when increasing $\beta$ further to $10^6$, there is a slight decline in Mean(PCK) to 88.05.

In summary, different combinations of hyperparameters yield different results, indicating that our method has great potential. It also demonstrates that both components of the loss function are essential for achieving good performance. Due to time constraints and limited computation resources, we did not make many tries and simply set $\alpha$ and $\beta$ to $10^{-3}$ and 1 in the main paper, respectively. Our follow-up work will analyze the hyperparameter in-depth, and we believe the KDSM can be further improved.

### A.6 DISCUSSION OF OVKD TASK FOR DIFFERENT SUPER-CATEGORIES

In order to evaluate the ability of KDSM to handle different super-categories with respect to the OVKD task, we divided the MP-78 dataset into two non-overlapping super-categories: the Face category and the Body category. As shown in Table 7, KDSM performs differently across these super-categories. Specifically, the Mean(PCK) values for the Face and Body categories are 81.89 and 73.97, respectively, indicating that KDSM performs better for the Face category compared to the Body category. The relatively lower performance for the Body category can be attributed to the greater variability in body poses. Overall, there may still be room for improvement in performance, especially for the Body category.

### A.7 FUTURE WORK

Our research focuses on achieving OVKD, a new and promising research topic, with satisfactory performance on regular scenes. Further improvement in challenging scenarios (e.g., occlusion, lighting, and resolution) will be left for our future work. Unlike traditional methods that rely on manual annotation, OVKD offers valuable recognition to arbitrary keypoints without prior annotation, especially for rare species and keypoint categories. In this paper, we follow the few-shot keypoint detection method (Xu et al., 2022a) to make evaluations on regular scenes. In addition, we add some detection results of our method under occlusion in Fig. 5 (a), and we can see that our method can solve some occlusion cases well. We also show some failure cases under occlusion in Fig. 5 (b).

