# OpenReview forum: "Language-driven Open-Vocabulary Keypoint Detection for Animal Body and Face"
_ICLR.cc/2024/Conference — ICLR 2024 Conference Withdrawn Submission_

### Official Review · Reviewer_eZk5 · 2023-10-18

**Soundness:** 3 good
**Presentation:** 3 good
**Contribution:** 3 good
**Rating:** 6
**Confidence:** 4

**Summary:**

This paper aims to build an open-vocabulary keypoint detector, using text prompts to localize arbitrary keypoints of any species. It is achieved by associating text prompt with relevant keypoint features, by mapping the textual features and detected keypoint heatmaps into a shared semantic space.
In order to adapt to species of different number of keypoints, all keypoint categories are clustered by kmeans into groups according to their textual embeddings.
The experiments on MP100 shows SoTA performance using a zero-shot fashion, even compared with few-shot keypoint detection.

**Strengths:**

With the proposed clustered keypoint categories (keypoint domain), the model can generalize to different species and various keypoint categories.

**Weaknesses:**

1. The keypoints that lack semantic information are not suitable for language-driven keypoint detector, such as keypoints on clothes and furniture. On the other hand, those keypoints are becoming increasingly important these days, espeicially for clothes.
2. Related to 1), the detector seems to only work for very sparse keypoints, lacking the ability to upscale the number of keypoints. For instance, the classical 68 facial landmarks.

**Questions:**

1. Since "animal species" are needed in the text prompt, how does this model generalize to very rare species (rare in CLIP)?
2. How is the performance compared with the semi-supervised methods, which also only needs 1 or 5 shots [A][B]? As they can deal with denser keypoints, I would recommend to at least add a discussion in the paper.

[A] Few-shot Geometry-Aware Keypoint Localization (CVPR2023)

[B] 3FabRec: Fast Few-shot Face alignment by Reconstruction (CVPR2020)

---

> ### Author Response · Authors · 2023-11-23
> **Response to eZk5**
>
> **Q1**:Since "animal species" are needed in the text prompt, how does this model generalize to very rare species (rare in CLIP)?
>
> **A1**: Thanks for your question. The OVKD task we proposed is a novel and promising task to the field of computer vision. As the pioneering work, there is still ongoing research to improve its performance. In the future, we plan to enhance the representation of keypoint categories to enable the detection of new species solely based on their keypoint category descriptions and species' appearance descriptions. Additionally, we would like to draw your attention to our unique contributions, such as introducing the OVKD task and incorporating special designs like Domain Distribution Matrix Matching and the Vision-keypoint relational Awareness Module.
>
> **Q2**:How is the performance compared with the semi-supervised methods, which also only needs 1 or 5 shots [A][B]? As they can deal with denser keypoints, I would recommend to at least add a discussion in the paper.
> [A] Few-shot Geometry-Aware Keypoint Localization (CVPR2023)
> [B] 3FabRec: Fast Few-shot Face alignment by Reconstruction (CVPR2020)
>
> **A2**: Thank you for your suggestion. Semi-supervised keypoint detection methods have entirely different technical approaches and motivations from the category-agnostic few-shot keypoint detection methods, such as the POMNet (Xu et al., 2022a), and the zero-shot OVKD task we propose in the paper. While semi-supervised methods may be capable of handling denser keypoints, our focus is on addressing the keypoint detection problem in a zero-shot category-agnostic manner.
> It is difficult to find a find a suitable approach to compare the semi-supervised methods with our method.
> So we will include a discussion on semi-supervised keypoint detection in the final version of the related work section.
>
> Xu et al., 2022a. Pose for everything: Towards category-agnostic pose estimation. ECCV 2022.

---

### Official Review · Reviewer_As86 · 2023-10-27

**Soundness:** 3 good
**Presentation:** 2 fair
**Contribution:** 2 fair
**Rating:** 5
**Confidence:** 4

**Summary:**

This paper presents a novel open-vocabulary keypoint detection framework, especially for recognizing animal keypoints. The problem and the motivation are clearly addressed and described, and experiments look promising. I agree that the open-vocabulary setting for the animal / human keypoint detection is promising and can contribute insights to the community.

**Strengths:**

This paper formulates novel frameworks for the OVKD task, including a baseline framework and an augmented one. Both frameworks show promising performance on different OVKT settings. In particular, the augmented version outperforms the baseline framework by a great margin. The studied problem is also an interesting topic in the community, and the motivation is well-written.

**Weaknesses:**

Although the introduction and motivation part are well-written, I find that the quality of the method description and experiments is not as good as I expected. In general, I have many confusions after reading the method descriptions and also have questions about the experiments. Please read the following section for specific questions. Overall, I think the unclear method makes the quality of this paper quite questionable.

**Questions:**

1. In Figure 2, I found that there is a 'Zero-shot' term on the top right. However, I cannot understand which part is zero-shot. In the paper, I found that the baseline method is still trained to deliver keypoint detection, except that the text_encoder is fixed. If the authors mean that the zero-shot can be carried on novel categories, it is quite misleading to put "zero-shot" on the overall baseline framework.
2. For the augmented framework, there are even more confusions. For example, the authors state that K-means is used to cluster the keypoint categories. However, I cannot understand: What are the actual data or features to be clustered? Why they should be clustered? Why K-means? Why not use other clustering algorithms?
3. In addition to question 2, I found a term called "binary domain distribution matrix". Still, I am confused about: What is binary domain distribution matrix? What's its formal definition? What does it represent? Why do we need binarized matrix?
4. Furthermore, why do we need to reorder channels for H'? What problem does the reordering solve?
5. In the experiment, I have the following question: The new MP-78 split is not clear: which species are used for training and which new species are used for testing? Will there possible be data leaking?
6. In Table 3, I found that VKLA improves a lot, while, according to my understanding, the so-called VKLA is just some extra attention modules that for sure can certainly improve performance. What if we stack more VKLS modules, does it further improve performance? If so, the contribution of this paper is quite questionable. Correct me if my understanding is wrong.
7. I have a question about the experiment design. Talking about OVKD, I suppose the proposed method should be working in different settings, such as pre-train keypoint knowledge only on human keypoints, and test on animal keypoints. In theory, I think the proposed should still be working. It would better to discuss this.
8. There are some unclear presentations (or typos?). In Figure 4, bottom 2 raws, what is 'keypint category'? Shouldn't that be some specific keypoint names?

---

> ### Author Response · Authors · 2023-11-23
> **Response to Reviewer As86 (Q1-Q3)**
>
> **Q1**: In Figure 2, I found that there is a 'Zero-shot' term on the top right. However, I cannot understand which part is zero-shot. In the paper, I found that the baseline method is still trained to deliver keypoint detection, except that the text\_encoder is fixed. If the authors mean that the zero-shot can be carried on novel categories, it is quite misleading to put "zero-shot" on the overall baseline framework.
>
> **A1**: Thank you for your question. In Figure 2, the 'Zero-shot' label signifies the baseline method's capability to generalize to novel keypoint categories without specific training on those categories. This is contributed to the text encoder from CLIP trained by large-scale image-text data and achieves generalization ability. So the term 'zero-shot' in the context of the baseline framework refers to its ability to handle unseen categories during testing.
>
> **Q2**: For the augmented framework, there are even more confusions. For example, the authors state that K-means is used to cluster the keypoint categories. However, I cannot understand: What are the actual data or features to be clustered? Why they should be clustered? Why K-means? Why not use other clustering algorithms?
>
> **A2**: Thank you for your comment.
>
> **A-What to be clustered**: As mentioned in Section 3.3, ``We perform K-means clustering on all the keypoint categories in the training set and cluster them into O groups according to their textual embeddings." The data we cluster is the textual embeddings (feature) of the keypoint descriptions obtained from the CLIP language model.
>
> **A-Why should be clustered**: During training, if we have 78 species and 15 keypoint categories per species, we would have 78*15 = 1170 (animal species, keypoint category) combinations. It is not practical to represent each combination with a separate channel of heatmaps. Instead, our goal is to represent multiple keypoint categories using a single channel of the groundtruth heatmaps.
> By dividing all the keypoint categories into several groups and learning keypoint categories in the same group together (where each group corresponds to one heatmap channel), we can optimize the training process and avoids wasting computational resources. Using this method, we can represent 1170 combinations using only 100 channels.
>
> **Why K-means**: There are various clustering algorithms available, and we chose the relatively simple K-means algorithm. It is worth noting that other clustering methods could also be used.
>
> **Q3-1**: In addition to question 2, I found a term called "binary domain distribution matrix". Still, I am confused about: What is binary domain distribution matrix? What's its formal definition? What does it represent?
>
> **A3-1**:
> After clustering, our objective is to assign each heatmap to a specific group (domin) representing combinations of (animal species, keypoint category). To accomplish this, we employ a binary domain distribution matrix that indicates the corresponding group (domin) for each $i$-th text prompt, thereby determining which heatmap corresponds to the $i$-th text prompt.
>
> As shown in Section 3.3, for every training sample, we pre-compute its binary domain distribution matrix $\mathbf{D} \in \mathbb{R}^{K\times O}$ (where $O=100$ in our implementation) based on the keypoint categories associated with that specific sample. Here, $K$ represents a fixed number that is greater than or equal to the maximum number of keypoints across all samples. The value of $\mathbf{D}{ij}$ is set to 1 only if the $i$-th keypoint belongs to the $j$-th group. In cases where the number of keypoints $K'$ for the sample is less than $K$, we assign $\mathbf{D}{ij} = 0$ for $i \in [K' + 1, K]$ and $j \in [1, O]$.
>
> In summary, if $\mathbf{D}_{ij} == 1$, it signifies that the $j$-th channel of the groundtruth heatmap corresponds to the $i$-th text prompt.
>
> **Q3-2**: Why do we need binarized matrix?
>
> **A3-2**: The use of a binarized matrix is essential for two main reasons:
>
> a) We compute the cross-entropy loss between the predicted distribution matrix and the annotated distribution matrix during training.
>
> b) The definition of the distribution matrix also requires it to be binary.

---

> ### Author Response · Authors · 2023-11-23
> **Response to Reviewer As86 (Q4-Q8)**
>
> **Q4**:Furthermore, why do we need to reorder channels for H'? What problem does the reordering solve?
>
> **A4**: Thank you for your question. Since the predicted heatmap ordering does not correspond to the prompt ordering, during testing, we need to determine the position $j$-th of the heatmap corresponding to the $i$-th prompt based on the predicted domain distribution matrix. Then, we reorder H' based on the domain distribution matrix. This ensures that the heatmaps are aligned correctly with their corresponding prompts. You can find a detailed process in Section 3.3.
>
>
> **Q5**: In the experiment, I have the following question: The new MP-78 split is not clear: which species are used for training and which new species are used for testing? Will there possible be data leaking?
>
> **A5**: Thank you for your question. In our experiment, we strictly follow the principles of "transductive generalized zero-shot learning" as defined by Pourpanah et al. (2022). The combinations of (animal species, keypoint category) in our dataset are unique and non-overlapping. We will provide publicly accessible splits along with the source code.
>
> Pourpanah et al., 2022. A review of generalized zero-shot learning methods. TPAMI (2022).
>
> **Q6**: In Table 3, I found that VKLA improves a lot, while, according to my understanding, the so-called VKLA is just some extra attention modules that for sure can certainly improve performance. What if we stack more VKLS modules, does it further improve performance? If so, the contribution of this paper is quite questionable. Correct me if my understanding is wrong.
>
> **A6**: Thank you for your question. In Table 4, we observe that increasing the number of self-attention blocks from 1 to 3 resulted in a noticeable performance improvement (row 1 vs. row 3). However, further increasing the number of self-attention blocks to 4 did not yield significant gains (row 3 vs. row 4). For the number of cross-attention blocks, we have a similar observation and choose to use three cross-attention blocks in our implementation.
> This suggests that there is a diminishing return in performance when stacking additional attention blocks beyond a certain point.
> Besides, it is important to note that the contribution of this paper extends beyond the VKLA module, as we also propose Domain Distribution Matrix Matching, and other special designs to build this first OVKD work.
>
>
> **Q7**: I have a question about the experiment design. Talking about OVKD, I suppose the proposed method should be working in different settings, such as pre-train keypoint knowledge only on human keypoints, and test on animal keypoints. In theory, I think the proposed should still be working. It would better to discuss this.
>
> **A7**: Thank you for your consideration. All combinations of (animal species, keypoint category) in our dataset are distinct and non-overlapping for both the training and testing sets. Additionally, we have employed a random split of the dataset into 5 splits for each experiment setting. This experimental design, akin to that of POMNet (Xu et al., 2022a), is deemed effective and fair.
> Our experiment settings encompass various scenarios, including the example where human keypoints are in the training set and some animal keypoints are in the testing set (Setting B, Split 1). These different settings allow us to evaluate the performance of the proposed method under different conditions. Also, beyond humans and animals, in our future work, we plan to extend the work to arbitrary object categories (e.g., cars, cloth, and furniture), which should be significant in the field of computer vision.
>
> Xu et al., 2022a. Pose for everything: Towards category-agnostic pose estimation. ECCV 2022.
>
> **Q8**: There are some unclear presentations (or typos?). In Figure 4, bottom 2 raws, what is 'keypint category'? Shouldn't that be some specific keypoint names?
>
> **A8**: Thank you for pointing that out. For the experiments involving new species, each new species has multiple keypoint categories. Due to the limitations of space in the paper, we couldn't list out all the specific keypoint prompts. Therefore, we used "{keypoint category}" as a placeholder to represent them. Additionally, there were indeed spelling mistakes in the word "keypoint" in Figure 4. We apologize for the oversight, and in the final version of the paper, we will correct the spelling error.

---

> > ### Comment · Reviewer_As86 · 2023-11-23
> >
> > Many thanks for the authors' answers. They clarify a lot. However, unfortunately, I believe this paper requires a major revision to meet my publication standard. So I will keep my rating.
> >
> > I think there will be a much higher chance that this paper can be accepted after comprehensive refinement to improve clarity and presentation.

---

### Official Review · Reviewer_r55N · 2023-10-30

**Soundness:** 3 good
**Presentation:** 3 good
**Contribution:** 3 good
**Rating:** 5
**Confidence:** 4

**Summary:**

The paper defines a new Open-Vocabulary Keypoint Detection (OVKD) task to locate arbitrary keypoints of any species. The authors propose a KDSM method to utilize visual and textual information for this task. In particular, KDSM contains a Vision-keypoint relational awareness module for visual and language information interaction, and a domain distribution matching module for semantic alignment and generalization. Experiments show the effectiveness of the method.

**Strengths:**

1. The defined OVKD task is important for the development of the keypoint detection task. It will be beneficial for more real-world applications and research.
2. The proposed domain distribution matrix matching module can process arbitrary keypoints and species input flexibly.
3. Experiments show that the proposed method outperforms the baseline by a large margin.

**Weaknesses:**

1. Unclear experimental settings:
In the definition of keypoint categories, will keypoints that are structurally consistent be categorized as a single keypoint category? For example, the front paw of a panda, a human's hand, and the forehooves of a horse. Will there be keypoints that are structurally consistent in both the training and test sets (such as the above examples)? If the keypoint categories in the training and test sets are structurally exclusive, (for instance, the train set lacks facial keypoints while the model is tested on facial keypoints; the train set lacks forelimbs keypoints while the model is tested on forelimb keypoints), how will the model perform?

2. The motivation should be further clarified:
In Section 3.3, it is suggested to provide more explanations to clarify why “The baseline framework struggle with generalization...performance” and why the proposed modules can address the problem.

3. Visualizations of the domain distribution matrix will help to understand the proposed method better. In addition, please provide some matched examples. For instance, given a prompt in the test set, please provide the training prompts falling in the same bin (in O groups). This can give us a more comprehensive understanding of your approach.

**Questions:**

See the weaknesses

---

> ### Author Response · Authors · 2023-11-23
> **Response to Reviewer r55N**
>
> **Q1**: Unclear experimental settings: In the definition of keypoint categories, will keypoints that are structurally consistent be categorized as a single keypoint category? For example, the front paw of a panda, a human's hand, and the forehooves of a horse. Will there be keypoints that are structurally consistent in both the training and test sets (such as the above examples)? If the keypoint categories in the training and test sets are structurally exclusive, (for instance, the train set lacks facial keypoints while the model is tested on facial keypoints; the train set lacks forelimbs keypoints while the model is tested on forelimb keypoints), how will the model perform?
>
> **A1**: Thank you for your question. We apologize for the previous lack of clarity in our paper. In our experiment settings, which strictly fall under the category of "transductive generalized zero-shot learning" (Pourpanah et al., 2022), the combinations of (animal species, keypoint category) are represented as "Keypoint Categories," and they are unique and non-overlapping. Even if there may be structural consistency among certain keypoints, each combination is treated as distinct and categorized separately. All splits will be made publicly accessible along with the source code.
>
>
> **Q2**: The motivation should be further clarified: In Section 3.3, it is suggested to provide more explanations to clarify why “The baseline framework struggles with generalization...performance” and why the proposed modules can address the problem.
>
> **A2**: Thank you for your question. The performance gap between the baseline and our KDSM, as shown in Tables 1 and 2, indicates the baseline approach's poor generalization ability. Additionally, the absence of the Vision-keypoint relational Awareness (VKLA) Module in the baseline hinders its effectiveness in capturing the relationship between textual and visual features. Furthermore, through the Domain Distribution Matrix Matching, we leverage knowledge from seen species and keypoints to improve the prediction of unseen keypoint categories in arbitrary species.
> In the final version of the paper, we will provide more comprehensive explanations in Section 3.3, supported by detailed experimental results, to provide a clearer understanding of the motivation behind our proposed modules and their effectiveness in addressing the challenges of generalization in the baseline framework.
>
> **Q3**:Visualizations of the domain distribution matrix will help to understand the proposed method better. In addition, please provide some matched examples. For instance, given a prompt in the test set, please provide the training prompts falling in the same bin (in O groups). This can give us a more comprehensive understanding of your approach.
>
> **A3**: Thank you for your suggestions. We will show some results of the domain distribution matrix in our final version, where we can observe that the predicted domain distribution matrix after training is close to the ground truth, demonstrating the effectiveness of the learning process.
> Regarding your second question, we provide you with two examples, as follows:
> Test prompt: "The back of a tern in the photo.", Training prompt: "The left shoulder of a panther body in the photo;
> Test prompt: "The belly of a tern in the photo.", Training prompt: "The abdomen of a vinegar fly in the photo."
> And we will show more in the final version of the paper.

---

### Official Review · Reviewer_yVmq · 2023-10-30

**Soundness:** 2 fair
**Presentation:** 3 good
**Contribution:** 2 fair
**Rating:** 3
**Confidence:** 4

**Summary:**

The work introduces the task of Language-Driven Open-Vocabulary Keypoint Detection (OVKD) for animal body and face keypoint localization. The existing methods for keypoint detection are limited to specific keypoints and species, requiring manual annotation or support images. OVKD aims to overcome these limitations by using text prompts to detect arbitrary keypoints of any species. The proposed framework, called Open-Vocabulary Keypoint Detection with Semantic-feature Matching (KDSM), leverages the relationship between text and vision using language models and domain distribution matrix matching. Extensive experiments show that KDSM outperforms the baseline framework and achieves impressive results in OVKD.

**Strengths:**

-	The authors introduce a new open-vocabulary task for animal keypoint detection.

-	The proposed framework improves the generalization capability across different species and keypoint categories, making it applicable to previously unseen keypoints and species.

-	KDSM integrates domain distribution matrix matching and some special designs to reinforce the relationship between language and vision.

**Weaknesses:**

-	This work lacks a comparison of complexities. The author should compare their approach with previous few-shot methods about the network parameters and inference time. Since this work employs CLIP, there might be a potential for a higher number of parameters which might lead to an unfair comparison.

-	Regarding CLAMP, although the author mentions that CLAMP uses the same keypoint categories, the technical methods between the two works exhibit minimal differences. Furthermore, the authors do not compare their results with CLAMP on the dataset.

-	What will the result be if both language and few-shot images are applied for training/inference simultaneously? Relying solely on language can be imprecise for species that have not been observed previously, making it challenging to generalize.

-	The authors only conduct experiments on the MP-78 dataset. Results on a single dataset can hardly be compelling.

**Questions:**

-	The comparison of complexity.
-	Comparisons with CLAMP.
-	The results of using both language and few-shot images.
-	More experiments on other datasets.

---

> ### Author Response · Authors · 2023-11-23
> **Response to Reviewer yVmq**
>
> **Q1**: This work lacks a comparison of complexities. The author should compare their approach with previous few-shot methods about the network parameters and inference time. Since this work employs CLIP, there might be a potential for a higher number of parameters which might lead to an unfair comparison.
>
> **A1**: Thank you for your question. It is indeed true that incorporating CLIP increases the number of parameters and inference time. However, we believe that the benefits of leveraging the language model to achieve OVKD outweigh the computational complexity it introduces. Our approach achieves zero-shot keypoint detection solely using text descriptions, without relying on support images, while still achieving performance comparable to few-shot methods. Therefore, we hope that you recognize the contributions made in proposing and implementing the OVKD task, as well as introducing the domain distribution matrix matching method and other task-specific designs.
>
> Additionally, CLIP has been extensively used in other open-vocabulary tasks and has been compared with few-shot methods in previous studies (e.g., [ref]).
>
>
> [ref] Boyi Li, Kilian Q Weinberger, Serge Belongie, Vladlen Koltun, and René Ranftl. Language-driven semantic segmentation. arXiv preprint arXiv:2201.03546, 2022.
>
> **Q2**: Regarding CLAMP, although the author mentions that CLAMP uses the same keypoint categories, the technical methods between the two works exhibit minimal differences. Furthermore, the authors do not compare their results with CLAMP on the dataset.
>
> **A2**: Thanks for your opinion. We have significantly different motivations and approaches than CLAMP. CLAMP is not an open vocabulary method and limits the language space to the same keypoint categories, which limits its ability to expand the focus from body keypoints to include facial keypoints or to be effective across species with different numbers of keypoint categories. Therefore, CLAMP cannot be applied to the MP-78 dataset.
>
> Differently, we introduce the OVKD task, and our method has many differences and special designs that CLAMP does not have, such as domain distribution matrix matching and visual keypoint relationship awareness modules. It is difficult to find a suitable way to compare the contrastive learning method CLAMP with our method. In addition, MP-78 dataset includes the AP-10K dataset used in CLAMP, which shows the generalization ability of our method to a certain extent.
>
> **Q3**: What will the result be if both language and few-shot images are applied for training/inference simultaneously? Relying solely on language can be imprecise for species that have not been observed previously, making it challenging to generalize.
>
> **A3**: Thank you for your question. Our motivation is to achieve zero-shot keypoint detection without relying on supporting images but only text descriptions. Therefore, our experimental setup does not involve using support images and language simultaneously for training/inference. We recognize its challenges, but it is also the goal of this work. As the pioneering work of this direction, we achieved some satisfactory results. For previously unobserved species, we hope that textual descriptions will help detect keypoints.
>
> Also, we thank you for your suggestion. In our future work, we plan to extend our work to utilize both text and images, which should improve its generalization ability.
>
> **Q4**: The authors only conduct experiments on the MP-78 dataset. Results on a single dataset can hardly be compelling.
>
> **A4**: Thank you for your concern. Compared with AP-10K and other datasets, the MP-78 dataset currently contains the most species categories and keypoint categories. MP-78 contains COCO, AFLW, MascaquePose, OneHand10K, AP-10K, AnimalWeb, CUB-200, etc. The diverse composition of MP-78 makes it the best choice to evaluate the performance of our approach across a variety of challenges and scenarios.

---

### Official Review · Reviewer_R5x4 · 2023-10-31

**Soundness:** 2 fair
**Presentation:** 3 good
**Contribution:** 2 fair
**Rating:** 6
**Confidence:** 5

**Summary:**

This paper proposes the open-vocabulary keypoint detection (OVKD) with the goal of detecting arbitrary keypoints of arbitray species from query image given text prompts. There is a baseline OVKD model and another advanced OVKD model proposed. Both models use CLIP as text encoder  while the ResNet or ViT as vision encoder. Moreover, the open-vocabulary Keypoint Detection with Semantic-feature Matching (KDSM) is proposed to leverage the benefit of language models, exploit the relationship between text and vision, and employ domain distribution matrix matching to enhance performance. The experiments are performed in a subset of MP100 and the results show the effectiveness of proposed OVKD model.

**Strengths:**

1. The task of OVKD is proposed to detect arbitrary keypoints.

2. Keypoint Detection with Semantic-feature Matching (KDSM) is proposed, which improves performance.

3. The experiments show the effectiveness of proposed method.

**Weaknesses:**

1. From the abstract, contributions, and conclusion, all appear "some special designs", what are these special designs?

2. When we check the comparison methods, it seems there is no method called FS-ULUS in paper [1]. Is there something wrong or just modifying their methods? It would be better to use the model name indicated in paper, for example, FSKD.

    [1] Few-shot keypoint detection with uncertainty learning for unseen species @ CVPR'22

3. Since there already exists an open-vocabulary keypoint detection work like CLAMP [2], what is the performance when comparing your method to CLAMP on Animal pose dataset in the setting of five-leave-one-out problem? Namely training on four species while testing on the left one species. The results of PCK @ 0.1 should be presented in paper for comparisons.

    [2] CLAMP+Prompt-based Contrastive Learning for Connecting Language and Animal Pose @ CVPR'23

4. The dimension of text feature (K=100, C=64) is only 64, which makes me doubt whether the model leverages the language prior or not.

5. KDSM essentially is matching and retrival. The global matching matrix will cause K-k invalid text prompts, which would waste computation.

6. Why not use CLIP's vision encoder, while using the ResNet50 pre-trained on imagenet. If the text & vision model do not match, their multi-modal embeddings are not aligned anymore. How can it claim that "leverage the language model's knowledge"?

7. PCK@0.2 may be too high to cause over-confident results.

8. What is the meaning of "query the names of datasets that do not have semantic naming"? How do you use chatgpt to query? Just manually query?

9. Moreover, I still doubt that zero-shot yields better performance than few-shot model, if images are not seen at training phase.

10. The split of seen and unseen keypoint categories should disclose for fair future comparisons. However, in appendix, one still cannot find the concrete splits.

**Questions:**

.

**Details Of Ethics Concerns:**

.

---

> ### Author Response · Authors · 2023-11-23
> **Response to Reviewer R5x4 (Q1-Q4)**
>
> **Q1**: From the abstract, contributions, and conclusion, all appear "some special designs", what are these special designs?
>
> **A1**: Thank you for your comment. Sorry for our unclear presentation. The special designs of our methods include the following:
>
> a) The Vision-keypoint Relation Awareness Module is adept at capturing the semantic relationship between visual features and textual prompts.
>
> b) The meticulous selection and design of fundamental modules within the KDSM framework, including the Text Encoder, Vision Encoder, Keypoint Adapter, Vision Head, and Vision Adapter. These modules serve as indispensable components for accomplishing and achieving good results on the OVKD task.
>
> c) The process of constructing text prompts, recognizing the significance of well-crafted prompts in establishing a seamless bridge between language and keypoint features.
>
> d) The MP-78 dataset is well organized and annotated, making it suitable for OVKD tasks.
>
> We will enhance the explanations of these special aspects in the final version of the paper.
>
> **Q2**: When we check the comparison methods, it seems there is no method called FS-ULUS in paper [1]. Is there something wrong or just modifying their methods? It would be better to use the model name indicated in paper, for example, FSKD.
> [1] Few-shot keypoint detection with uncertainty learning for unseen species
>
> **A2**: We appreciate your careful review, and we apologize for any confusion. Few-shot Keypoint Detection (FSKD) represents a task concept, encompassing various FSKD methods like MAML, Fine-Tune, POMNet, and others. Consequently, we did not explicitly mark the method [1] as FSKD. Instead, we referred to the method "**F**ew-**s**hot keypoint detection with **u**ncertainty **l**earning for **u**nseen **s**pecies" as FS-ULUS.
> We will mark this naming in the final version.
>
> **Q3**: Since there already exists an open-vocabulary keypoint detection work like CLAMP [2], what is the performance when comparing your method to CLAMP on Animal pose dataset in the setting of five-leave-one-out problem? Namely training on four species while testing on the left one species. The results of PCK @ 0.1 should be presented in paper for comparisons.
>
> [2] CLAMP+Prompt-based Contrastive Learning for Connecting Language and Animal Pose @ CVPR'23
>
> **A3**: Thanks for your comments. CLAMP limits the language space to the same keypoint categories, so CLAMP is not an OVKD method. CLAMP has a limited ability to extend the focus from body keypoints to include facial keypoints or generalize effectively across species with varying numbers of keypoint categories. Consequently, applying CLAMP to the MP-78 dataset can not lead to successful results. It is difficult to find a suitable approach to compare the contrastive learning method, CLAMP, with our method. Furthermore, it is worth mentioning that the MP-78 dataset includes the AP-10K dataset used in CLAMP, which to some extent indicates the generalization capabilities of our method.
>
> Thanks for your comment about the PCK evaluation metric. We follow previous works (e.g., Xu et al., 2022a) to use PCK@0.2 while PCK@0.1 differs slightly regarding the metric threshold. We re-test some previously trained models using PCK@0.1, and the results are presented in the table below.
> | Framework | Shot   | Split1 | Split2 | Split3 | Split4 | Split5 | Mean  | Split1 | Split2 | Split3 | Split4 |
> |-----------|--------|--------|--------|--------|--------|--------|-------|--------|--------|--------|--------|
> | P-0.2     | P-0.2  | P-0.2  | P-0.2  | P-0.2  | P-0.2  | P-0.1  | P-0.1 | P-0.1  | P-0.1  | P-0.1  | P-0.1  |
> |FS-ULUS [1] | 1-shot | 73.69  | 70.65  | 63.97  | 71.14  | 63.65  | 68.62 | 55.05  | 54.93  | 45.79  | 50.85  |
> |FS-ULUS [1] | 5-shot | 78.34  | 79.67  | 76.89  | 81.52  | 75.23  | 78.33 | 62.66  | 67.65  | 56.37  | 67.16  |
> | Baseline  | 0-shot | 56.06  | 55.36  | 54.35  | 53.07  | 50.66  | 53.90 | 44.12  | 42.07  | 41.95  | 41.27  |
> | KDSM      | 0-shot | 84.02  | 87.99  | 83.22  | 83.20  | 80.25  | 83.14 | 72.72  | 77.22  | 70.21  | 69.87  |
>
> It is worth noting that since (Xu et al., 2022a) exclusively employ PCK@0.2, we do not retest it using PCK@0.1. It is clear that results on both PCK@0.2 and PCK@0.1 can demonstrate our effectiveness. We will add PCK@0.1 results into the final version.
>
> [1] Few-shot keypoint detection with uncertainty learning for unseen species
> Xu et al., 2022a. Pose for everything: Towards category-agnostic pose estimation. ECCV 2022.
>
> **Q4**:The dimension of text feature (K=100, C=64) is only 64, which makes me doubt whether the model leverages the language prior or not.
>
> **A4**: Thank you for your question. We use the text features extracted from the language model provided by CLIP, whose dimension is 512. For better interaction with image features, we compress it to 64, while maintaining most of the original information.

---

> ### Author Response · Authors · 2023-11-23
> **Response to Reviewer R5x4 (Q5-Q10)**
>
> **Q5**: KDSM essentially is matching and retrival. The global matching matrix will cause K-k invalid text prompts, which would waste computation.
>
> **A5**: Thank you for your attention. Despite the presence of K-k invalid text prompts, the computation of the matrix-matching operation is very low against the feature extraction, and we only need a one-time invalid text feature extraction.
>
> **Q6**: Why not use CLIP's vision encoder, while using the ResNet50 pre-trained on imagenet. If the text and vision model do not match, their multi-modal embeddings are not aligned anymore. How can it claim that "leverage the language model's knowledge"?
>
> **A6**: Thanks for your concerns. Following previous research (Ni et al., 2022), we trained a task-specific visual encoder instead of fine-tuning the original CLIP visual encoder, but we both still leverage the language model's knowledge (that is why we can achieve OVKD). The table below demonstrated that the ResNet50-based visual encoder outperformed the CLIP pre-trained ViT-B/32 model. This is because OVKD involves handling diverse pose variations and joint localization, which requires dense region-level features, while the CLIP visual encoder learns the global image-level features.
>
> | Text Encoder                | Vision Encoder  | Split1 | Split2 | Split3 | Split4 | Split5 | Mean(PCK) |
> |-----------------------------|-----------------|--------|--------|--------|--------|--------|-----------|
> | Paired with ViT-B/32 (CLIP) | ViT-B/32 (CLIP) | 52.54  | 49.25  | 59.23  | 50.30  | 45.17  | 51.30     |
> | Paired with ViT-B/32 (CLIP) | ResNet50        | 79.02  | 71.35  | 76.68  | 79.79  | 74.67  | 76.90     |
>
> Ni et al., Expanding language-image pretrained models for general video recognition. ECCV 2022.
>
> **Q7**:PCK@0.2 may be too high to cause over-confident results.
>
> **A7**: Thank you for your concern. We have answered this question in **A3**.
>
> **Q8**: What is the meaning of "query the names of datasets that do not have semantic naming"? How do you use chatgpt to query? Just manually query?
>
> **A8**: We apologize for the confusion in our previous statement. For keypoint types like clothing that lack semantic naming, we have removed such data in MP-100. For "querying the names of datasets that do not have semantic naming," we are referring to keypoint types that do have semantic meaning, but we may not be able to provide a precise definition or description for them. In such cases, we use ChatGPT to query and obtain the names of these keypoints.
>
> For example, we use a query like "How to anatomically describe the second joint of the index finger?" to obtain the name of a specific keypoint. All these queries are performed manually, and then we build the dataset MP-78. We will refine this statement in the final version of the paper.
>
> **Q9**: Moreover, I still doubt that zero-shot yields better performance than few-shot model, if images are not seen at training phase.
>
> **A9**: Thank you for your inquiry. In some language-driven open-vocabulary approaches, superior performance compared to some few-shot methods has been demonstrated across various tasks, as seen in open-vocabulary semantic segmentation [3]. Additionally, In addition, few-shot keypoint detection methods like POMNet (Xu et al., 2022a) do not have access to images of new categories during training, but can only utilize supporting images of new categories during testing. Therefore, we think it is reasonable for zero-shot methods to exhibit better performance than few-shot methods.
>
> Xu et al., 2022a. Pose for everything: Towards category-agnostic pose estimation. ECCV 2022.
> [3] Boyi Li, Kilian Q Weinberger, Serge Belongie, Vladlen Koltun, and René Ranftl. Language-driven semantic segmentation. arXiv preprint arXiv:2201.03546, 2022.
>
> **Q10**: The split of seen and unseen keypoint categories should disclose for fair future comparisons. However, in appendix, one still cannot find the concrete splits.
>
> **A10**: Thank you for your question. Our experiment settings strictly fall under the category of "transductive generalized zero-shot learning" (Pourpanah et al., 2022). In our paper, the combinations of (animal species, keypoint category) are represented as "Keypoint Categories," and they are both unique and non-overlapping. All splits will be made publicly accessible along with the source code.
>
> Pourpanah et al., 2022. A review of generalized zero-shot learning methods. TPAMI (2022).